# A Novel Prognostic DNA Methylation Panel for Colorectal Cancer

**DOI:** 10.3390/ijms20194672

**Published:** 2019-09-20

**Authors:** Hsin-Hua Chung, Chih-Chi Kuo, Cheng-Wen Hsiao, Chao-Yang Chen, Je-Ming Hu, Chih-Hsiung Hsu, Yu-Ching Chou, Ya-Wen Lin, Yu-Lueng Shih

**Affiliations:** 1Graduate Institute of Medical Sciences, National Defense Medical Center, No. 161, Sec. 6, Minquan East Rd., Neihu District, Taipei 11490, Taiwan; 2Teaching and Research Office, Tri-Service General Hospital Songshan Branch, No. 131, Jiankang Rd., Songshan District, Taipei 10581, Taiwan; 3Division of Colorectal Surgery, Department of Surgery, Tri-Service General Hospital, National Defense Medical Center, No. 325, Sec. 2, Chenggong Rd., Neihu District, Taipei 11490, Taiwan; 4Adjunct Instructor, School of Medicine, National Defense Medical Center, No. 161, Sec. 6, Minquan East Rd., Neihu District, Taipei 11490, Taiwan; 5Teaching Office, Tri-Service General Hospital, No. 325, Sec. 2, Chenggong Rd., Neihu District, Taipei 11490, Taiwan; 6School of Public Health, National Defense Medical Center, No. 161, Sec. 6, Minquan East Rd., Neihu District, Taipei 11490, Taiwan; 7Department and Graduate Institute of Microbiology and Immunology, National Defense Medical Center, No. 161, Sec. 6, Minquan East Rd., Neihu District, Taipei 11490, Taiwan; 8Graduate Institute of Life Sciences, National Defense Medical Center, No. 161, Sec. 6, Minquan East Rd., Neihu District, Taipei 11490, Taiwan; 9Division of Gastroenterology, Department of Internal Medicine, Tri-Service General Hospital, National Defense Medical Center, No. 325, Sec. 2, Chenggong Rd., Neihu District, Taipei 11490, Taiwan

**Keywords:** colorectal cancer (CRC), NK6 homeobox 1 (*NKX6.1*), LIM homeobox transcription factor 1α (*LMX1A*), sex-determining region Y-box 1 (*SOX1*), zinc finger protein 177 (*ZNF177*), DNA methylation

## Abstract

Colorectal cancer (CRC) is one of the most common cancers and the second leading cause of cancer-related deaths. Discrepancies in clinical outcomes are observed even among patients with same-stage CRC due to molecular heterogeneity. Thus, biomarkers for predicting prognosis in CRC patients are urgently needed. We previously demonstrated that stage II CRC patients with *NKX6.1* methylation had poor 5-year overall survival. However, the methylation frequency of *NKX6.1* was only 23% in 151 pairs of CRC tissues. Thus, we aimed to develop a more robust prognostic panel for CRC using *NKX6.1* in combination with three genes: *LIM homeobox transcription factor 1α* (*LMX1A*), *sex-determining region Y-box 1* (*SOX1*), and *zinc finger protein 177* (*ZNF177*). Through quantitative methylation analysis, we found that *LMX1A*, *SOX1*, and *ZNF177* were hypermethylated in CRC tissues. *LMX1A* methylation was significantly associated with poor 5-year overall, and disease-free survivals in stage I and II CRC patients. Sensitivity and specificity analyses of the four-gene combination revealed the best sensitivity and optimal specificity. Moreover, patients with the four-gene methylation profile exhibited poorer disease-free survival than those without methylation. A significant effect of the four-gene methylation status on overall survival and disease-free survival was observed in early stage I and II CRC patients (*p* = 0.0016 and *p* = 0.0230, respectively). Taken together, these results demonstrate that the combination of the methylation statuses of *NKX6.1*, *LMX1A*, *SOX1*, and *ZNF177* creates a novel prognostic panel that could be considered a molecular marker for outcomes in CRC patients.

## 1. Introduction

Colorectal cancer (CRC) is the second leading cause of cancer-related deaths globally, following lung cancer, and was responsible for an estimated 862,000 deaths in 2018 [1]. With advances in strategies for early detection and treatment, the survival of CRC patients has improved over the last two decades. However, the 5-year survival rate of CRC patients is only 65%, due to the high rates of recurrence and metastasis [2].

The outcomes of CRC patients are closely related to the tumor stages at diagnosis [3]. The 5-year relative survival rate of patients with localized, i.e., stage I, CRC is 90%, whereas that of patients with cancer spreading to the surrounding tissues or regional lymph nodes (stage II and III) is approximately 70%. In addition, the 5-year survival rate of CRC patients in whom the cancer has spread to distant sites (stage IV) is unfortunately <15%. While the data indicate that most stage II or III CRC patients might have good outcomes, a small percentage of these patients remains at risk of recurrence and death due to metastatic disease. Therefore, the identification of molecular markers, other than the cancer stage, is necessary for predicting prognosis; moreover, it may lead to the development of effective diagnostic, therapeutic, and preventive strategies for CRC.

DNA methylation is an important epigenetic modification of cancer cells, which is considered to be one of the major mechanisms of the inactivation of tumor-related suppressor genes that ultimately contribute to carcinogenesis [4,5]. In addition to its role in repressing gene expression by influencing the chromatin structure and interfering with transcription initiation, many studies demonstrated the clinical utility of DNA methylation biomarkers in various cancers, including CRC [6,7]. For CRC screening, a stool-based screening test for *N-Myc downstream-regulated gene 4* (*NDRG4*) and *bone morphogenic protein 3* (*BMP3*) methylation, as well as a blood-based assay for *septin 9* (*SEPT9*) methylation, were approved by the United States Food and Drug Administration [8,9,10,11]. However, markers based on abnormal DNA methylation patterns remain poor choices for diagnosis, prognostic prediction, and treatment selection; and the identification of novel genes in CRC is highly desired.

We recently demonstrated the abnormal methylation of *NK6 homeobox 1* (*NKX6.1*) in CRC and found that *NKX6.1* methylation was an independent indicator of 5-year disease-free survival in stage II CRC patients receiving adjuvant chemotherapy [12]. However, the methylation frequency of *NKX6.1* was only 23% in our cohort of 151 pairs of CRC tissues. Therefore, a DNA methylation panel might potentially improve the sensitivity of detection. Several genes, including *LIM homeobox transcription factor 1α* (*LMX1A*), *sex-determining region Y-box 1* (*SOX1*), and *zinc finger protein 177* (*ZNF177*) were previously identified to be methylated in a high fraction of cancers, and the methylations of genes alone or in combination were shown to be potential biomarkers for cancers [13,14,15,16,17,18,19,20,21,22,23,24]. However, the methylation frequencies of *LMX1A*, *SOX1*, and *ZNF177*, as well as the potential of their combined methylation status to be a prognostic factor of CRC, are unknown. In the current study, we aimed to determine the methylation frequencies of *LMX1A*, *SOX1*, and *ZNF177* in CRC and to elucidate the prognostic utility and efficacy of their methylation status in combination with the *NKX6.1* methylation status in CRC.

Our analysis of the methylation levels of *LMX1A*, *SOX1*, and *ZN177* in CRC revealed that the combination of the methylation statuses of *NKX6.1*, *LMX1A*, *SOX1*, and *ZNF177* was an independent indicator of the outcomes of CRC patients. These results indicate that the combination of four-gene methylation statuses could be a novel prognostic DNA methylation marker for CRC.

## 2. Results

### 2.1. Abnormal Methylation of LMX1A, SOX1, and ZNF177 in CRC

To investigate whether *LMX1A*, *SOX1*, and *ZNF177* were aberrantly methylated in CRC, we used the data of the Infinium Human Methylation 450K BeadChips from the MethHC database to analyze the methylation levels of *LMX1A* (NM_001174069), *SOX1* (NM_005986), and *ZNF177* (NM_003451) in 369 tissue samples from colon and rectal adenocarcinoma patients (Figure 1A). The analyses showed that the average β values were significantly higher in the tumor group than in the normal control group for all three genes (*p* < 0.0001). To further confirm the abnormal methylation phenotype, we examined the methylation status of *LMX1A*, *SOX1*, and *ZNF177* in five CRC cell lines (HCT8, HCT116, HT29, SW480, and SW620) using the MSP assay (Figure 1B) and quantified the methylation levels using the Q-MSP assay (Figure 1C). The results of both the gel-based MSP assay and the Q-MPS assay revealed that *LMX1A*, *SOX1*, and *ZNF177* were heavily methylated in over half of the cell lines. To analyze the correlation between *LMX1A*, *SOX1*, and *ZNF177* and their gene expressions, we treated HCT116 cells with a DNA methylation inhibitor (5′-aza-2′-deoxycytidine; DAC) alone or in combination with an HDCA (histone deacetylase) inhibitor (Trichostatin A; TSA). The reverse transcription polymerase chain reaction (RT-PCR) data showed that the expressions of *LMX1A*, *SOX1*, and *ZNF177* were restored in HCT116 cells after the cells were treated with the two drugs (Figure 1B, right panel). The Q-MSP results showed that the DNA methylation levels (methylation indexes) of *LMX1A*, *SOX1*, and *ZNF177* were decreased after the HCT116 cells were treated with DAC only or the combination of DAC and TSA (Figure 1C). Then we quantified the methylation levels of *LMX1A*, *SOX1*, and *ZNF177* in 151 pairs of CRC tissues and found that the DNA methylation levels of *LMX1A*, *SOX1*, and *ZNF177* in 151 CRC tissue samples were significantly higher than those in the corresponding nontumor tissue samples (Figure 1D). Furthermore, we performed a receiver operating characteristic (ROC) curve analysis to discriminate between the CRC tissue samples and their nontumor counterparts (supporting information, Appendix A) and determined that the methylation frequencies of *LMX1A*, *SOX1*, and *ZNF177* were 22.5%, 13.9%, and 12.6%, respectively, under the best cut-off values (Table 1). 

### 2.2. The Association of LMX1A Methylation and Patient Survival

To elucidate whether *LMX1A*, *SOX1*, and *ZNF177* methylation could have a potential use in clinical practice, we explored the association of the methylation of *LMX1A*, *SOX1*, and *ZNF177* and the clinical characteristics of 151 CRC patients (Table 2). We found a statistically significant association between *LMX1A* methylation and tumor size (*p* = 0.0048) but did not observe significant associations between *SOX1* or *ZNF177* methylation and any of the clinicopathological parameters. To further investigate the relevance of gene methylation with survival, we next employed the Kaplan-Meier method with the log-rank test for each gene methylation. As shown in Figure 2A, the methylation of neither *LMX1A*, *SOX1*, nor *ZNF177* had any effect on the overall 5-year overall survival or disease-free survival of CRC patients. However, we found that early-stage (stage I or II) CRC patients with *LMX1A* methylation exhibited a poorer 5-year overall survival (*p* = 0.0108) and disease-free survival (*p* = 0.0468) than those without *LMX1A* methylation (Figure 2B).

### 2.3. The Efficacy of a Novel DNA Methylation Panel for Predicting the Prognosis of CRC

We previously demonstrated that *NKX6.1* was a novel prognostic biomarker for CRC and that the frequency of *NKX6.1* methylation was 23% in a cohort of 151 CRC tissues [12]. Here, we determined whether the methylation status of *NKX6.1*, in combination with those of *LMX1A*, *SOX1*, and *ZNF177*, had a greater power in detecting CRC. The sensitivity and specificity of two and three-gene combinations ranged from 21.9% to 37.7% and 97.4% to 98.7%, respectively, and the four-gene panel provided the best sensitivity, 39.7%, and the optimal specificity, 97.4% (Table 1). 

To further evaluate the utility of the four-gene methylation panel, we first analyzed its association with the clinicopathological characteristics. As shown in Table 3, the four-gene methylation panel was significantly associated with age (*p* = 0.0083), tumor size (*p* = 0.0323), and survival (*p* = 0.0297). To determine its prognostic utility, we next used the Cox proportional hazards model to identify independent factors associated with survival (Table 4). Univariate analysis revealed that the four-gene methylation panel was indeed a statistically significant indicator of overall survival (hazard ratio (HR), 2.95; 95% confidence interval (CI), 1.36–6.39; *p* < 0.01). In the multivariate analysis, gene methylation and chemotherapy were confirmed as independent factors for overall survival (HR for gene methylation, 3.43; 95%CI, 1.37–8.57; *p* < 0.01) (HR for chemotherapy, 0.32; 95%CI, 0.12–0.85; *p* < 0.05). Furthermore, the Kaplan–Meier survival analysis determined the effect of the four-gene methylation panel on survival, revealing that patients who exhibited methylation for all four genes had a worse 5-year overall survival (*p* = 0.0213) and disease-free survival (*p* = 0.0134) than those without methylation of any of the four genes (Figure 3a). Finally, the four-gene methylation status was associated with 5-year overall survival and disease-free survival in early-stage patients (Figure 3b; *p* = 0.0016 and *p* = 0.0230), and in stage II and III CRC patients (Figure 3c; *p* = 0.0009 and *p* = 0.0363).

## 3. Discussion 

The tumor-node-metastasis (TNM) staging is a key determinant of the therapy selection and prognosis of CRC patients. However, beyond creating a more complex tumor classification, the TNM staging system has not provided clinicians with the optimal treatment and management strategies that it was originally designed for. Recent advances in the determination of the molecular CRC subtypes have led to the identification of several prognostic biomarkers, based on molecular heterogeneity, which might contribute to the risk stratification of patients with the same disease stages and who are receiving the same treatment [25,26]. In the current study, we analyzed the methylation levels of *LMX1A*, *SOX1*, and *ZNF177* to elucidate novel changes in the methylation levels of genes in CRC and determined that the novel four-gene methylation panel, which includes *LMX1A*, *SOX1*, *ZNF177*, and *NKX6.1*, could discriminate the outcomes of CRC patients.

As in other cancers, CRC encompasses multiple molecular subtypes with specific characteristics. The CpG island methylator phenotype (CIMP) is one famous CRC subtypes, including tumors with a high frequency of hypermethylation at CpG islands [27]. CIMP-associated CRC has distinct epidemiology, histology, precursor lesions, and molecular features (e.g., *BRAF* and *KRAS* mutations) [28,29].

In addition, recent studies have reported that the prognosis was significantly poor in CIMP-high patients among the microsatellite stability (MSS) subgroup, although there was a trend of increased cancer-specific survival in CIMP-positive CRC patients [30,31,32,33,34,35,36]. However, the association of CIMP with the prognosis and survival of CRC subtypes remains unknown.

We previously reported several DNA methylation markers in various cancers, including cervical cancer, ovarian cancer, and hepatocellular carcinoma [13,16,19]. However, the utility of these markers or marker panels for CRC is unknown. Therefore, we designed a strategy to first verify the DNA methylation levels of specific markers that we identified in colon and rectal adenocarcinomas using array data from existing databases and then validated these results by a quantitative DNA methylation analysis of potential genes in the study cohort using Q-MSP. With this strategy, we recently reported *NKX6.1* hypermethylation as a new biomarker for the prediction of outcomes and therapy selection in stage II CRC patients [12]. In the current study, we not only showed that *LMX1A* hypermethylation was significantly associated with tumor size and poor 5-year overall survival rates and disease-free survival rates in early-stage CRC patients, but also demonstrated that the four-gene methylation panel, including *NKX6.1*, *LMX1A*, *SOX1*, and *ZNF177*, was a novel and independent prognostic tool for stage I and II, or stage II and III CRCs.

The four-gene methylation status was decreased and the expression level was restored in HCT116 cells after the cells were treated with DAC alone or with the combination of DAC/TSA. Moreover, the correlation analysis of the differential methylation and expression levels of *LMX1A*, *SOX1*, *ZNF177* and *NKX6.1* in colon adenocarcinoma patients was based on data derived from the MethHC database (supporting information in Appendix A). The inverse correlation between the gene expression and DNA methylation of *SOX1* (correlation = −0.2971, *p* < 0.0001) was statistically significant. There was no correlation between the methylation and gene expression of *LMX1A* (correlation = −0.00675) and *NKX6.1* (correlation = −0.1046) from the MethHC database. However, a positive correlation was observed between the gene expression level and the methylation status of *ZNF177* (correlation = 0.3487, *p* < 0.0001) (Appendix A). There are two *ZNF177* variants reported at the NCBI, but the probe was designed only for the variant NM_003451′s methylation analysis from MethHC. These controversial data may be due to the different variants of ZNF177 and the heterogeneity of the clinical samples. In summary, there was an inverse correlation between the methylation and expression of the four-gene panel in HCT116 cells. Among the markers, *LMX1A* is a LIM homeobox-containing gene, with roles in a wide variety of developmental contexts [37]; *SOX1* encodes a protein that plays a key regulatory role in neural cell fate determination and differentiation [38,39]; and *ZNF177* belongs to a zinc finger gene family encoding a large number of common transcription factors [40,41]. Previously, in addition to identifying DNA methylation biomarkers, we also revealed that *LMX1A* and *SOX1* were tumor suppressor genes in cervical cancer and hepatocellular carcinoma [42,43,44]. Additionally, the promoter-hypermethylations of *LMX1A*, *SOX1*, and *ZNF177* were previously reported in lung, stomach, and bladder cancers, respectively, by other groups [14,17,18]. In the two most recent reports, *SOX1* hypermethylation was also identified in colon cancer and CRC [45,46]. Overall, the data from previous studies, as well as the data from the current report, indicate promoter hypermethylation of these genes are common epigenetic events in multiple cancer types. The biological function of these four genes in CRC needs further investigation.

Despite the promising results, the current study does have limitations. The cutoff value for each gene was based on a hospital-based, retrospective case-control study from a research platform and may not be applied directly to clinical settings with larger populations. To determine its prognostic utility, we used the Cox proportional hazards model to identify independent factors associated with survival (Appendix A). The univariate analysis revealed that *LMX1A* and *NKX6.1* methylation was indeed a statistically significant indicator of overall survival (hazard ratio (HR) for *LMX1A* methylation, 2.58; *p* < 0.05) (HR for *NKX6.1* methylation, 2.60; *p* < 0.01). In the multivariate analysis, *NKX6.1* methylation, stage and chemotherapy were confirmed as independent factors for overall survival (HR for *NKX6.1* methylation, 6.06; *p* < 0.01) (HR for stage, 3.77; *p* < 0.05) (HR for chemotherapy, 0.26; *p* < 0.05). *LMX1A* methylation and chemotherapy were confirmed as independent factors for overall survival (HR for *LMX1A* methylation; *p* < 0.05) (HR for chemotherapy, 0.33; *p* < 0.05). However, our previous study demonstrated that the methylation frequency of *NKX6.1* was 23% in our cohort of 151 pairs of CRC tissues. In this study, we aimed to develop a DNA methylation panel that might potentially improve the sensitivity of the application. While the methylation of *SOX1* and *ZNF177* was not a statistically significant indicator of overall survival, the combination of these two genes could improve the sensitivity (23% to 39.7%) when discriminating 151 CRC tissues from nontumor tissues. Therefore, we used a four-gene methylation panel to elucidate its prognostic utility for CRC. As there was no patient survival information in the MethHC database, we could not determine whether the four-gene set had a prognostic value in these cohorts. In this study, we did not have any other available cohort to further validate the prognostic value. However, we analyzed data from PRECOG (https://precog.stanford.edu), a database for querying associations between gene expressions and clinical outcomes, to examine the association between the expression of the four-gene panel and the survival of CRC patients. A GEO dataset (GSE12945) showed that patients with a low *NKX6.1* expression had a poorer survival rate, compared with those with a high *NKX6.1* expression (*p* = 0.0159; Appendix A). For the other three genes, *LMX1A*, *SOX1*, and *ZNF177*, there was no significant correlation between the gene expression and patient survival. The reason for these results was due to the limited patient survival information and the heterogeneity of the clinical samples. In summary, this is a pilot study that indicates that the abnormal DNA methylation panel is a potential prognostic biomarker of CRC. Nonetheless, the prognostic value of the four-gene panel needs another independent cohort to further validate these results, and an in vitro validation of the biological function of these genes will provide a better understanding of the findings.

The molecular heterogeneity of CRC may aid in determining clinical survival of CRC patients. The current study examined the prognostic value of the methylation levels of *LMX1A*, *SOX1*, and *ZNF177* in CRC patients. The results suggest that the combination of *NKX6.1*, *LMX1A*, *SOX1*, and *ZNF177* promoter methylation statuses is a potentially novel stage-independent prognostic marker.

## 4. Materials and Methods 

### 4.1. MethHC Database

MethHC (http://MethHC.mbc.nctu.edu.tw) is a web resource specifically focused on the DNA methylation of human cancers [13]. In the current study, we used DNA methylation data in the MethHC database for preliminary analysis of the methylation levels of *LMX1A*, *SOX1*, and *ZNF177* in 46 tissue samples from normal individuals and 369 tissue samples from colon or rectal adenocarcinoma patients.

### 4.2. Clinical Samples and Cell Lines

Our cohort included 151 paired tumor tissue and adjacent nontumor tissue specimens that were obtained from the Tri-Service General Hospital. The use of these samples was approved by our Institutional Review Board (TSGHIRB number: 098-05-292). The clinicopathologic characteristics of the patients were summarized previously [12]. Additionally, five CRC cell lines (HCT8, HCT116, HT29, SW480, and SW620) were used in the current study. HT29 cells were purchased from the American Type Culture Collection. HCT8, HCT116, SW480, and SW620 cells were purchased from the Food Industry Research and Development Institute (Taiwan). 

### 4.3. DNA Methylation and Gene Expression Analysis

For DNA methylation analysis, the genomic DNA of clinical samples and cell lines were extracted and bisulfite-converted as previously described [12]. CpG methylated human genomic DNA (Thermo Fisher Scientific, San Diego, CA, USA) and DNA extracted from normal peripheral blood lymphocytes were modified by sodium bisulfite to generate positive and negative controls, respectively. MS-PCR and Q-MSP were performed as previously described. For Q-MSP, the DNA methylation levels were assessed by determining the methylation index (MI) using the following formula: 100 × 2^−((Cp of Gene)−(Cp of COL2A))^. The Q-MSP primer sequences used in the current study were as follows: *LMX1A*-F tgggacgcgggattgtaaattttat, *LMX1A*-R aaaccctcgaaacgtctctacaaaa, *SOX1*-F ggttgtatcgtaatcgttttttgtaggtt, *SOX1*-R cctccaactcgaaaactacaacttct, *ZNF177*-F ggaagtgggcgttcgtcgtttc, *ZNF177*-R cccttccctcccgattccg, *COL2A*-F gggaagatgggatagaagggaatat, and *COL2A*-R tctaacaattataaactccaaccaccaa. For gene expression analysis, reverse transcription polymerase chain reaction (RT-PCR) was conducted as previously described [12]. The RT-PCR primer sequences used in the current study are shown in the Appendix A.

### 4.4. Statistical Analysis

Statistical analyses were performed using GraphPad Prism software (version 4.03; GraphPad Software, La Jolla, CA, USA) and SPSS software (IBM SPSS Statistics 21; Asia Analytics Taiwan, Taipei, Taiwan). The Mann–Whitney *U* and Wilcoxon signed-rank tests were used to determine differences between gene methylation levels and disease status. Receiver operating characteristic (ROC) curves were generated to calculate the cut-off values for *LMX1A*, *SOX1*, and *ZNF177* methylation for discriminating tumors from nontumor tissues. χ^2^ test and an χ^2^ test for trend were used to calculate the associations between gene methylations and clinical parameters. Kaplan–Meier curves were calculated to estimate survival rates at five years after treatment. The log-rank test was used to compare the association between survival and gene methylation. The Cox proportional hazards model was used to identify factors for overall survival.

## 5. Conclusions

In summary, our results demonstrate that the combination of the methylation statuses of *NKX6.1*, *LMX1A*, *SOX1*, and *ZNF177* is a novel prognostic panel that could be considered a molecular marker for outcomes in CRC patients. This is a pilot study that indicates that the abnormal DNA methylation panel is a potential prognostic biomarker of CRC. Nonetheless, the prognostic value of the four-gene panel needs another independent cohort to further validate these results.

## Figures and Tables

**Figure 1 ijms-20-04672-f001:**
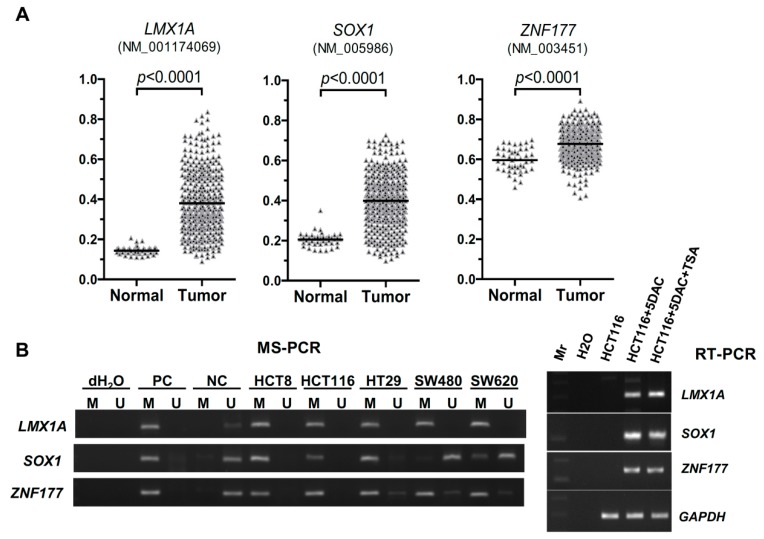
DNA methylation levels of *LIM homeobox transcription factor 1α* (*LMX1A*), *sex-determining region Y-box 1* (*SOX1*), and *zinc finger protein 177* (*ZNF177*) in colorectal carcinoma (CRC). (**A**) DNA methylation array data for *LMX1A*, *SOX1*, and *ZNF177* in 45 tissue samples from normal individuals and 369 tissue samples from colon and rectal adenocarcinoma patients from the MethHC database are shown. The results are represented as average (AVG) β values for the probes. Black lines indicate the mean of AVG β value. The *p*-values for *LMX1A*, *SOX1*, and *ZNF177* methylation levels among the groups (normal versus tumor) were determined by the Mann–Whitney *U* test. (**B**) The promoter methylation statuses of *LMX1A*, *SOX1*, and *ZNF177* in five CRC cell lines were analyzed by MSP with methylated and unmethylated-specific primers. PC: positive control; NC: negative control. Gene expression levels of three genes and the internal reference *GAPDH* in HCT116 cells treated with 5 µM DAC or 5 µM DAC combined with 300 nM TSA (5DAC/TSA), or left untreated, were analyzed by reverse transcription polymerase chain reaction (RT-PCR). Mr: molecular marker, or DNA ladder. (**C**) Quantitative DNA methylation levels of *LMX1A*, *SOX1*, and *ZNF177* in five CRC cell lines were analyzed by Q-MSP. Quantitative DNA methylation levels of *LMX1A*, *SOX1*, and *ZNF177* in HCT116 cells treated with 5 µM DAC or 5 µM DAC combined with 300 nM TSA (5DAC/TSA), or left untreated, were determined by Q-MSP. The results are represented as differences in the methylation index (MI). (**D**) Quantitative DNA methylation levels of *LMX1A*, *SOX1*, and *ZNF177* were determined in 151 paired CRC tissue samples and the adjacent nontumor tissue samples (NT) by Q-MSP. The results are represented as differences in the methylation index. Black lines indicate mean methylation index. *p* values for methylation levels among the groups were determined by the Wilcoxon signed-rank test.

**Figure 2 ijms-20-04672-f002:**
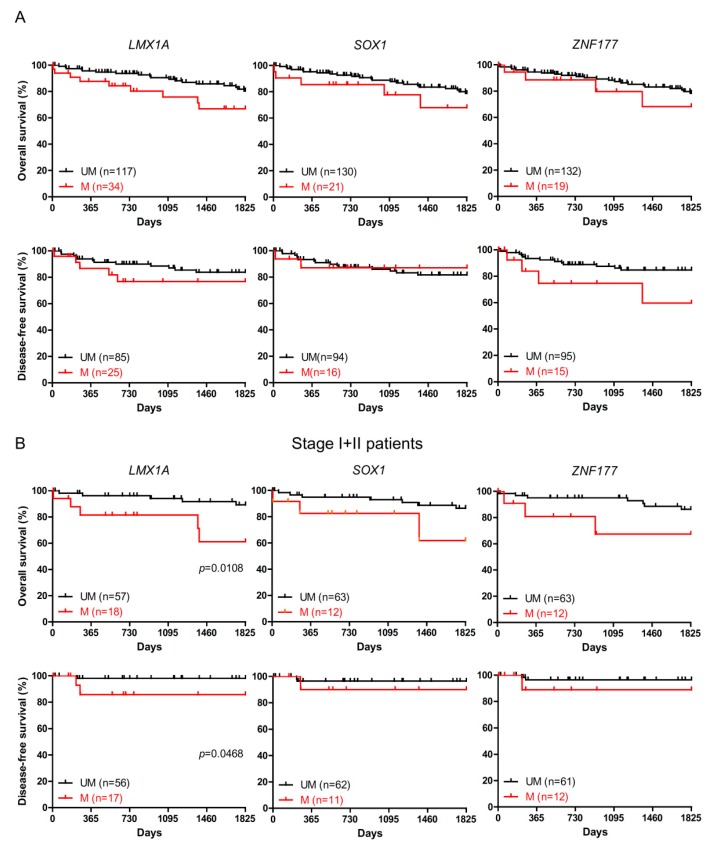
Correlation analysis between *LMX1A*, *SOX1*, and *ZNF177* methylation levels and survival in CRC patients. (**A**) The 5-year overall survival and disease-free survival rates of CRC patients with different *LMX1A*, *SOX1*, and *ZNF177* methylation statuses are presented. Red lines indicate cases with *LMX1A* methylation (methylation index (MI) > 16.84), *SOX1* methylation (MI > 30.26), and *ZNF177* methylation (MI > 51.14). Black lines indicate cases without *LMX1A* methylation (MI ≤ 16.84), *SOX1* methylation (MI ≤ 30.26), and *ZNF177* methylation (MI ≤ 51.14). (**B**) The 5-year overall survival and disease-free survival rates of stage I and II CRC patients.

**Figure 3 ijms-20-04672-f003:**
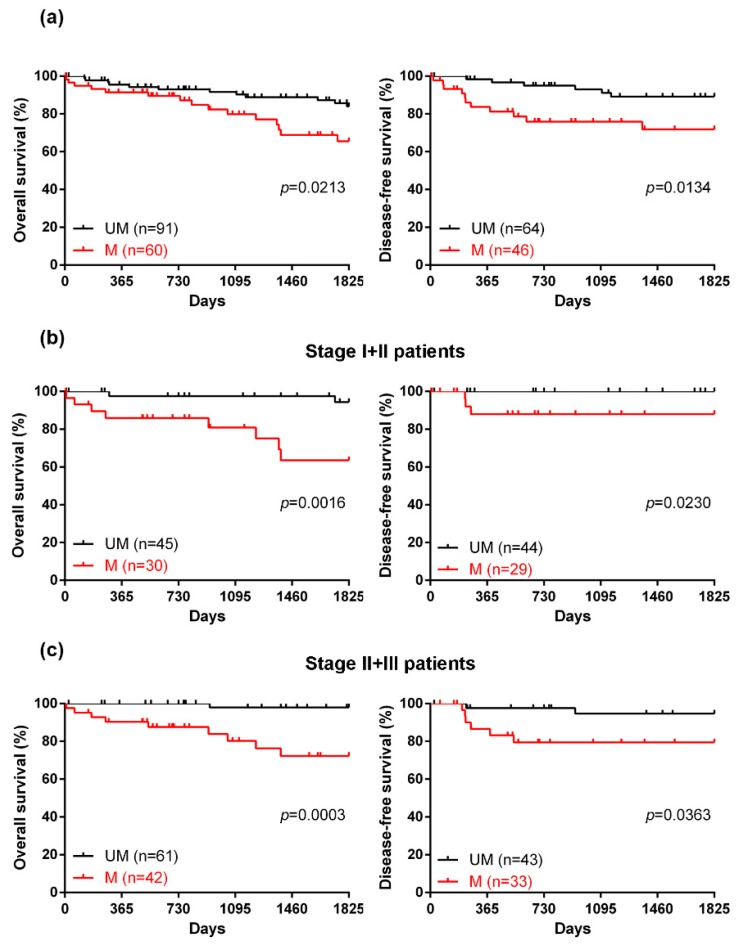
Correlation analysis between the four-gene methylation panel and the survival of CRC patients. (**a**) The 5-year overall survival and disease-free survival rates of CRC patients according to the methylation status are presented. Red line: methylated cases discriminated by the four-gene methylation panel; black line: unmethylated cases. (**b**) The 5-year overall survival and disease-free survival rates of stage I and II CRC patients. (**c**) The 5-year overall survival and disease-free survival rates of stage II and III CRC patients.

**Table 1 ijms-20-04672-t001:** Sensitivity and specificity of candidate gene sets for discriminating 151 CRC from nontumor tissues.

Gene Set	Best Cut-Off Value	Sensitivity	Specificity
*NKX6.1*	MI > 18.42	23.2%	99.3%
*LMX1A*	MI > 16.84	22.5%	99.3%
*SOX1*	MI > 30.26	13.9%	98.7%
*ZNF177*	MI > 51.14	12.6%	99.3%
*NKX6.1* or *ZNF177*		33.1%	98.7%
*NKX6.1* or *LMX1A*		31.1%	98.7%
*LMX1A* or *ZNF177*		30.5%	98.7%
*NKX6.1* or *SOX1*		29.1%	98.0%
*LMX1A* or *SOX1*		27.2%	98.0%
*SOX1* or *ZNF177*		21.9%	98.7%
*NKX6.1* or *LMX1A* or *ZNF177*		37.7%	98.0%
*NKX6.1* or *SOX1* or *ZNF177*		36.4%	98.0%
*LMX1A* or *SOX1* or *ZNF177*		33.8%	98.0%
*NKX6.1* or *LMX1A* or *SOX1*		33.8%	97.4%
*NKX6.1* or *LMX1A* or *SOX1* or *ZNF177*		39.7%	97.4%

The methylation level of each gene was assessed as a methylation index (MI).

**Table 2 ijms-20-04672-t002:** Association between gene methylation and clinicopathological characteristics in 151 CRC patients.

Symbol		*LMX1A*			*SOX1*			*ZNF177*		
		(MI > 16.84)			(MI > 30.26)			(MI > 51.14)		
		UM	M	*p*	UM	M	*p*	UM	M	*p*
Age (63.44 ± 14.46)	<64 years	62	12	0.0508	67	10	0.8162	69	8	0.4673
	≥64 years	55	22		63	11		63	11	
Sex	Female	61	18	1.0000	66	13	0.3597	70	9	0.8067
	Male	56	16		64	8		62	10	
Stage	І	20	2	0.2885	22	0	0.0510	18	4	0.5910
	ІІ	38	15		41	12		45	8	
	ІІІ	40	10		43	7		46	4	
	ІV	19	7		24	2		23	3	
Tumor grade	Well differentiated	4	0	0.5387	4	0	0.6122	4	0	0.1327
	Moderately differentiated	95	27		104	18		104	18	
	Poorly or undifferentiated	15	5		18	2		20	0	
	Missing data	4	1		4	1		4	1	
Tumor size	≤5 cm	72	10	0.0048	72	10	0.6151	74	8	0.5894
	>5 cm	37	18		46	9		48	7	
	Missing data	9	5		12	2		10	4	
No. of lymph node	≥12	95	22	0.3755	101	16	1.0000	105	12	0.2676
	0–11	15	6		18	3		17	4	
	Missing data	8	5		11	2		10	3	
Chemotherapy	No	31	8	1.0000	32	7	0.4141	34	5	0.7729
	Yes	79	20		87	12		88	11	
	Missing data	8	5		11	2		10	3	
Recurrence	No	74	19	0.5483	79	14	0.8095	82	11	0.8025
	Yes	43	15		51	7		50	8	
Survival	Alive	100	24	0.0722	108	16	0.5381	109	15	0.7494
	Dead	17	10		22	5		23	4	

Unmethylation (UM) of each gene is represented as MI ≤ the best cut-off value; methylation (M) of each gene is represented as MI > the best cut-off value.

**Table 3 ijms-20-04672-t003:** The associations between the four-gene methylation panel and clinicopathological characteristics in 151 CRC patients.

Symbol		*NKX6.1* or *LMX1A* or *SOX1* or *ZNF177*		
		UM	M	*p*
Age (63.44 ± 14.46)	<64 years	52	21	0.0083
	≥64 years	39	39	
Sex	Female	48	31	1.0000
	Male	43	29	
Stage	І	16	6	0.3678
	ІІ	30	23	
	ІІІ	32	18	
	ІV	13	13	
Tumor grade	Well differentiated	4	0	0.2355
	Moderately differentiated	72	50	
	Poorly or undifferentiated	13	7	
	Missing data	4	1	
Tumor size	≤5 cm	57	25	0.0323
	>5 cm	28	27	
	Missing data	7	7	
No. of lymph node	≥12	74	43	0.4653
	0–11	11	10	
	Missing data	7	6	
Chemotherapy	No	24	15	1.0000
	Yes	61	38	
	Missing data	7	6	
Recurrence	No	59	34	0.3928
	Yes	32	26	
Survival	Alive	80	44	0.0297
	Dead	11	16	

Unmethylation (UM) of each gene is represented as MI ≤ the best cut-off value; methylation (M) of each gene is represented as MI > the best cut-off value.

**Table 4 ijms-20-04672-t004:** Univariate and multivariate analysis of overall survival using clinical characteristics and the the four-gene methylation panel in 151 CRC patients.

Variable	Univariate Analysis Hazard Ratio	Multivariate Analysis Hazard Ratio
	(95% Confidence Interval)	(95% Confidence Interval)
Age (years)	0.99 (0.96–1.02)	0.98 (0.95–1.01)
Sex (female versus male)	1.05 (0.49–2.22)	1.35 (0.57–3.21)
Gene methylation		
Unmethylation	Reference	Reference
Methylation	**2.95 (1.36–6.39) ****	**3.43 (1.37–8.57) ****
Stage		
I + II	Reference	Reference
III + IV	1.77 (0.81–3.87)	2.66 (0.98–7.26)
Tumor grade		
Well + moderately	Reference	Reference
Poorly or undifferentiated	0.67 (0.16–2.86)	0.74 (0.16–3.36)
Tumor size		
≤5 cm	Reference	Reference
>5 cm	0.97 (0.42–2.21)	0.63 (0.25–1.61)
No. of lymph node		
≥12	Reference	Reference
0–11	0.42 (0.17–1.07)	0.49 (0.17–1.40)
Chemotherapy		
No	Reference	Reference
Yes	0.51 (0.23–1.13)	**0.32 (0.12–0.85) ***

* *p* < 0.05; ** *p* < 0.01.

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
