# Peer review of "A Novel Prognostic DNA Methylation Panel for Colorectal Cancer"

_ijms, 2019, doi:10.3390/ijms20194672_

Round 1
Reviewer 1 Report
No further revisions required.
Author Response
We greatly appreciate the positive comments from you.
Reviewer 2 Report
The authors addressed most of my concerns. There are still some minor points:
Line 118: "Furthermore, we In addition, we performed" needs to be corrected
Line 250: "The inverse correlation..." . This point needs to be corrected. A correlation coefficient of -0.00675 or -0.1046 means that there is no correlation at all even if the p-value is significant. Thus, interpretation of the correlation data is not correct.
Author Response
Response:
We appreciate for your comments and suggestions. We agree with your opinion and correct these mistakes in our revised manuscript.
line 118: We have deleted ‘We In addition” in our revised manuscript.
line 250. We have corrected this interpretation of the correlation data “The inverse correlation between the gene expression and DNA methylation of SOX1 (correlation=-0.2971, p<0.0001) was statistically significant. There was no correlation between the methylation and gene expression of LMX1A (correlation=-0.00675) and NKX6.1 (correlation=-0.1046) from the MethHC database.”
Moreover, we have used a professional English editing service to check the revised manuscript.
Reviewer 3 Report
Re-Revision
A Novel Prognostic DNA Methylation Panel for Colorectal Cancer
Hsin-Hua Chung, Chih-Chi Kuo, Cheng-Wen Hsiao, Chao-Yang Chen, Je-Ming Hu, Chih-Hsiung Hsu, Yu-Ching Chou, Ya-Wen Lin, Yu-Lueng Shih
In this revision of the earlier version, authors have made attempt with the HDAC inhibitor treatment and validate the results on the cell line. That is encouraging.
Limitations of the study has ben discussed, that is needed.
Verification of DNA methylation levels of specific markers and their correlation with the patient prognosis is the basis of this manuscript. The current cohort (used in the study) follows in line of these studies. Can author comment/verify its validity on any other cohort. This needs to be clearly specified in the writeup whether this identified prognostic signature is valid for specific cohort or to other cohorts also.
Line 110; HDCA inhibitor should be HDAC (Histone DeAcetylase) inhibitor
Line 136: Mr:DNA Marker: What dies it means?
Line 139: 5 μM 5DAC (Typo?)
Author Response
In this revision of the earlier version, authors have made attempt with the HDAC inhibitor treatment and validate the results on the cell line. That is encouraging.
Limitations of the study has been discussed, that is needed.
Response:
We greatly appreciate the positive comments from you.
Verification of DNA methylation levels of specific markers and their correlation with the patient prognosis is the basis of this manuscript. The current cohort (used in the study) follows in line of these studies. Can author comment/verify its validity on any other cohort. This needs to be clearly specified in the write-up whether this identified prognostic signature is valid for specific cohort or to other cohorts also.
Line 110; HDCA inhibitor should be HDAC (Histone DeAcetylase) inhibitor
Line 136: Mr:DNA Marker: What dies it means?
Line 139: 5 μM 5DAC (Typo?)
Response:
We appreciate for your comments and suggestions. We agree with your opinion and correct these mistakes in our revised manuscript.
We clearly specify this prognostic signature is only verified in our current cohort in the revised manuscript.
Line 288: “However, these results were only verified in our cohort and all patients included in the current study were Asian. Other cohort study is needed to validate these results.”
Line 110; HDCA inhibitor was corrected to HDAC (Histone Deacetylase) inhibitor
Line 136: Mr: it means molecular marker; Mr: Molecular marker, DNA ladder
Line 139: “5 μM 5DAC” was corrected to “5 μM DAC”
This manuscript is a resubmission of an earlier submission. The following is a list of the peer review reports and author responses from that submission.
Round 1
Reviewer 1 Report
This manuscript looks at the performance of a new 4 gene methylation panel as a prognostic marker. Overall it was well written and very clear. My major concern is that their new tool is not compared with current gold standards for predicting patient survival. As well limitations of the study should be better described at the end of the discussion
Minor concerns: Line 45 starting with "The significant effect of ......" was not clear to me I think it was comparing the 4 gene signature and the LMX1a gene but needs to be rewritten
Author Response
Please see the attachment
Reviewer 1
This manuscript looks at the performance of a new 4 gene methylation panel as a prognostic marker. Overall it was well written and very clear.
Response:
We greatly appreciate the positive comments from you.
My major concern is that their new tool is not compared with current gold standards for predicting patient survival. As well limitations of the study should be better described at the end of the discussion.
Response:
We appreciate for your comments and suggestions. We agree with your opinion and discuss this limitation of this study in “Discussion”. We have added the new sentence in our revised manuscript (Line 238, Page 10).
TNM staging is a key determinant of therapy selection and prognosis in CRC patients. However, beyond creating a more complex tumor classification, the TNM staging system has not provided clinicians with optimal treatment and management strategies that it was originally designed for. Recent advances in determination of the molecular CRC subtypes led to THE identification of several prognostic biomarkers based on molecular heterogeneity that might contribute to risk stratification of patients with the same disease stage receiving the same treatmen. In the current study, we analyzed the methylation frequencies of LMX1A, SOX1, and ZNF177 to elucidate novel changes in methylation of genes in CRC and determined that the novel four-gene methylation panel including LMX1A, SOX1, ZNF177, and NKX6.1 could discriminate outcomes in CRC patients. In summary, this is a pilot study that implies abnormal NKX6.1, LMX1A, SOX1 and ZNF177 methylation is a potential prognostic biomarker in CRC. In the future, we will use an independent cohort to further validate the clinical value of this marker and design in vitro and animal experiments to verify the effects of methylation and gene expression of these genes on CRC.
Minor concerns: Line 45 starting with "The significant effect of ......" was not clear to me. I think it was comparing the 4 gene signature and the LMX1a gene but needs to be rewritten.
Response:
We appreciate for your suggestion. We have replaced the new sentence in our revised manuscript (Line 45, Page 1)
Reviewer 2 Report
In the manuscript entitled "A Novel Prognostic DNA Methylation Panel for Colorectal Cancer" by Chung et al., the authors describe a panel of 4 methylation markers (NKX6.1, LMX1A, SOX1 and ZNF177) with prognostic value in colorectal cancer patients. Methylation data from 369 CRC patients were obtained from the MethHC database. In addition, 5 CRC cell lines and 151 primary CRC and matching non-malignant tissue samples were analysed for methylation of the 4 genes by MSP/qMSP assays. Methylation of LMX1A, SOX1 or ZNF177 was not prognostically relevant in CRC when analysed separately. However, CRC patients who exhibited methylation for all four genes had a worse 5-year overall survival and disease-free survival than those without methylation of any of the four genes. Overall, the data are of interest and the manuscript is well written.
Comments:
1. DNA methylation is closely associated with transcriptional gene expression. How good is the correlation between methylation of NKX6.1, LMX1A, SOX1 and ZNF177 and expression of these genes in the CRC patients? Is the combination of NKX6.1, LMX1A, SOX1 and ZNF177 expression also prognostically relevant?
2. Based on Table 3, the authors defined a tumor as methylated when NKX6.1 or LMX1A or SOX1 or ZNF177 were found to be methylated. It may be more robust to calculate a 4-gene methylation score using multivariate Cox regression (i.e. sum of the products of cox coefficients and methylation values) followed by maximally selected rank statistics to calculate the optimal cut-point for patient stratification.
Author Response
Please see the attachment
Reviewer 2
In the manuscript entitled "A Novel Prognostic DNA Methylation Panel for Colorectal Cancer" by Chung et al., the authors describe a panel of 4 methylation markers (NKX6.1, LMX1A, SOX1 and ZNF177) with prognostic value in colorectal cancer patients. Methylation data from 369 CRC patients were obtained from the MethHC database. In addition, 5 CRC cell lines and 151 primary CRC and matching non-malignant tissue samples were analysed for methylation of the 4 genes by MSP/qMSP assays. Methylation of LMX1A, SOX1 or ZNF177 was not prognostically relevant in CRC when analysed separately. However, CRC patients who exhibited methylation for all four genes had a worse 5-year overall survival and disease-free survival than those without methylation of any of the four genes. Overall, the data are of interest and the manuscript is well written.
Response:
We greatly appreciate the positive comments from you.
Comments:
1. DNA methylation is closely associated with transcriptional gene expression. How good is the correlation between methylation of NKX6.1, LMX1A, SOX1 and ZNF177 and expression of these genes in the CRC patients? Is the combination of NKX6.1, LMX1A, SOX1 and ZNF177 expression also prognostically relevant?
Response:
We appreciate for your comments and suggestions. To correlate the gene expression level and methylation status of NKX6.1, LMX1A, SOX1 and ZNF177 gene, we used the data of the Infinium Human Methylation 450K BeadChips and IlluminaHiSeq gene expression RNAseq from the TCGA database to analyze the methylation levels and gene expression of NKX6.1, LMX1A, SOX1 and ZNF177 in 394 tissue samples from colon and rectal adenocarcinoma patients. The analyses showed that the correlation between the gene expression RNAseq and average β values of LMX1A and ZNF177 was significantly negative correlation (Pearson correlation coefficient = -0.102, P < 0.05) and significantly positive correlation (Pearson correlation coefficient = 0.424, P < 0.01), but the NKX6.1 and SOX1 have no significantly correlation. In previous studies demonstrated a lot of genes have been showed an altered methylation and gene expression pattern between normal and tumor that is either progressing or maintained in carcinogenesis [1,2,3]. The TCGA databases showed the correlation between methylation and gene expression in normal tissues and tumor. However, we showed the methylation staus in 151 primary CRC and matching non-malignant tissue samples. Our results may represent a premalignant tissue condition in our cohort of 151 pairs of CRC tissues.
Reference
1. El Taghdouini, A., et al., Genome-wide analysis of DNA methylation and gene expression patterns in purified, uncultured human liver cells and activated hepatic stellate cells. Oncotarget, 2015. 6(29): p. 26729-45.
2. Ammerpohl, O., et al., Distinct DNA methylation patterns in cirrhotic liver and hepatocellular carcinoma. Int J Cancer, 2012. 130(6): p. 1319-28.
3. Hlady, R.A., et al., Epigenetic signatures of alcohol abuse and hepatitis infection during human hepatocarcinogenesis. Oncotarget, 2014. 5(19): p. 9425-43.
2. Based on Table 3, the authors defined a tumor as methylated when NKX6.1 or LMX1A or SOX1 or ZNF177 were found to be methylated. It may be more robust to calculate a 4-gene methylation score using multivariate Cox regression (i.e. sum of the products of cox coefficients and methylation values) followed by maximally selected rank statistics to calculate the optimal cut-point for patient stratification.
Response:
We thank for your question. Refer to your suggestion, we used multivariate Cox regression whether the methylation score of NKX6.1 in combination with those of LMX1A, SOX1, and ZNF177 and calculated the optimal cut-point. The four-gene panel provided the optimal sensitivity of 21.9% and the optimal specificity of 99.3%. Although, the specificity is better than our preliminary data (97.4%) but the sensitivity is decreased (from 39.7% to 21.9%). However, our previously study demonstrated the methylation frequency of NKX6.1 was 23% in our cohort of 151 pairs of CRC tissues. Thus, we want to develop a DNA methylation panel might potentially improve the sensitivity for detection. In the current study, we determine the methylation frequencies of LMX1A, SOX1, and ZNF177 in CRC and to elucidate the prognostic utility and efficacy of their methylation status in combination with the NKX6.1 methylation status in CRC. Therefore, we believe NKX6.1 in combination with LMX1A, SOX1, and ZNF177 is a more robust prognostic panel for CRC.
Reviewer 3 Report
The authors of this study present a prognostic indicator for colorectal cancer through the methylation of a four-gene panel (NKX6.1, SOX1, LMX1A and ZNF177). They concluded that the methylation status is a significant indicator of reduced survival in CRC patients.They previously observed this prognostic value in NKX6.1, however due to it only occurring in 23% of CRC patients, they need more to strengthen the tool. They successfully showed a combination of 4 genes LMX1A, NKX6.1, ZNF177, and SOX1 to be a predictive marker of CRC.
PROS
The paper is well written and provides a well-designed study.
The data clearly shows the effect of all four genes, when used in a single panel significantly reduces both overall and disease free in CRC patients up to stage III.
Overall the data is consistent and supports itself within the confines of the study.
They provide a well thought out conclusion to their findings and do not overreach.
CONS
Major
· One area that is not well defined is the specific effect of each individual gene in the 4-gene panel. Although earlier data suggest that the individual genes do not have a significant effect, it is unclear in this manuscript what impact each gene (or other combinations) is having in the group. For instance, Table 4 indicates a significant effect for the 4-gene panel on survival, however it is unclear based on the results of this study which of the 4 genes is imparting this effect.
· Although specific gene variants are discussed in Figure 1, their effect on various clinical parameters do not appear to be investigated in this study. It is unclear at first glance which variant(s) of these genes is being used to generate these clinical associations. And how (if at all) they would differ.
Minor
In vitro validation into the oncogenic consequences of the methylation of these genes will give a better understanding of the findings.
· Line 226, page 9 of 14: change "...and ZNF177 was an novel and independent prognostic tool..." to "...and ZNF177 was a novel and independent prognostic tool..."
Some values (e.g., Table 2: Effect of LMX1A based on age; Effect of SOX1 based on cancer staging) were near the border of statistical significance. If possible, repeating this type of experiment with a larger cohort of patients may indicate whether some of the genes being investigated could truly be significant or not.
Author Response
Please see the attachment
Reviewer 3
The authors of this study present a prognostic indicator for colorectal cancer through the methylation of a four-gene panel (NKX6.1, SOX1, LMX1A and ZNF177). They concluded that the methylation status is a significant indicator of reduced survival in CRC patients.They previously observed this prognostic value in NKX6.1, however due to it only occurring in 23% of CRC patients, they need more to strengthen the tool. They successfully showed a combination of 4 genes LMX1A, NKX6.1, ZNF177, and SOX1 to be a predictive marker of CRC.
PROS
The paper is well written and provides a well-designed study. The data clearly shows the effect of all four genes, when used in a single panel significantly reduces both overall and disease free in CRC patients up to stage III. Overall the data is consistent and supports itself within the confines of the study. They provide a well thought out conclusion to their findings and do not overreach.
Response:
We greatly appreciate the positive comments from you.
CONS
Major
One area that is not well defined is the specific effect of each individual gene in the 4-gene panel. Although earlier data suggest that the individual genes do not have a significant effect, it is unclear in this manuscript what impact each gene (or other combinations) is having in the group. For instance, Table 4 indicates a significant effect for the 4-gene panel on survival, however it is unclear based on the results of this study which of the 4 genes is imparting this effect.
Although specific gene variants are discussed in Figure 1, their effect on various clinical parameters do not appear to be investigated in this study. It is unclear at first glance which variant(s) of these genes is being used to generate these clinical associations. And how (if at all) they would differ.
Response:
We appreciate for your comments and suggestions. To determine its prognostic utility, we used the Cox proportional hazards model to identify independent factors associated with survival. Univariate analysis revealed that the NKX6.1 and LMX1A methylation were indeed a statistically significant indicator of overall survival (hazard ratio [HR] for NKX6.1 methylation, 2.60; 95% confidence interval [CI], 1.18–5.72; P < 0.01) (HR for LMX1A methylation, 2.58; 95% confidence interval [CI], 1.18–5.65; P < 0.05). In multivariate analysis, NKX6.1 methylation, stage and chemotherapy were confirmed as independent factors for overall survival (HR for NKX6.1 methylation, 6.06; 95%CI, 2.18–16.88; P < 0.01) (HR for stage, 3.77; 95%CI, 1.25–11.36; P < 0.05) (HR for chemotherapy, 0.26; 95%CI, 0.09–0.73; P < 0.05). And LMX1A methylation and chemotherapy were confirmed as independent factors for overall survival (HR for LMX1A methylation, 3.30; 95%CI, 1.25–8.76; P < 0.05) (HR for chemotherapy, 0.33; 95%CI, 0.12–0.87; P < 0.05). However, our previously study demonstrated the methylation frequency of NKX6.1 was 23% in our cohort of 151 pairs of CRC tissues. Thus, we want to develop a DNA methylation panel might potentially improve the sensitivity for detection. Although SOX1 and ZNF177 methylation were not statistically significant indicator of overall survival, these two genes were added which improve the sensitivity (23% to 39.7%) for discriminating 151 CRC from nontumor tissues.
Minor
In vitro validation into the oncogenic consequences of the methylation of these genes will give a better understanding of the findings.
Response:
We appreciate for the comments and suggestions of the reviewers. We agree with your opinion. We will design in vitro and animal experiments to verify the effects of methylation and gene expression of these genes on CRC in the future.
Line 226, page 9 of 14: change "...and ZNF177 was an novel and independent prognostic tool..." to "...and ZNF177 was a novel and independent prognostic tool..."
Response:
We are sorry for the typing error. We have corrected it in our revised manuscript (Line 225, Page 9).
Some values (e.g., Table 2: Effect of LMX1A based on age; Effect of SOX1 based on cancer staging) were near the border of statistical significance. If possible, repeating this type of experiment with a larger cohort of patients may indicate whether some of the genes being investigated could truly be significant or not.
Round 2
Reviewer 3 Report
Revision
A Novel Prognostic DNA Methylation Panel for Colorectal Cancer
Hsin-Hua Chung, Chih-Chi Kuo, Cheng-Wen Hsiao, Chao-Yang Chen, Je-Ming Hu, Chih-Hsiung Hsu, Yu-Ching Chou, Ya-Wen Lin, Yu-Lueng Shih
In this revision of the earlier version, authors have not made much attempt to modify the discussion or the results section in which the earlier comment could have been answered, regarding the individual impact of the genes.
They have changed the location of few lines here and there, and the invitro validation suggested for future studies.
The concern “Although specific gene variants are discussed in Figure 1, their effect on various clinical parameters do not appear to be investigated in this study. It is unclear at first glance which variant(s) of these genes is being used to generate these clinical associations. And how (if at all) they would differ” has not been answered.
Since the authors used the cell lines to show the methylation status of the genes, RTPCR from the same cell lines, can verify the methylation effect on the transcription. More methylation, less RNA. That will be a functional proof that methylation of these genes in the cell lines have any outcome at the cell line level, this can be performed for this study. No need for the future paper. Demethylating agents are commercially available, using them can successfully show, that these genes are truly methylated in the cell lines and the effect can be reverted.
The writeup need to be checked for proper sentence framing at multiple places.